# Exploring the key points and key shots in table tennis matches based on survival analysis

Muzi Li[1,2], Qing Yang[1]*

1 School of Physical Education, Soochow University, Suzhou, Jiangsu, China, 2 Tianping College Nanjing Campus Public Teaching Department, Suzhou University of Science and Technology, Nanjing, Jiangsu, China

* qyang@suda.edu.cn

## Abstract

This study aimed to investigate the Key Points (KP) and Key Shots (KS) in singles matches for elite table tennis players, enhancing the knowledge of winning patterns in table tennis. A total of 60 men's singles and 64 women's singles matches were analyzed, all data sourced from events such as the World Championships, World Cup, Olympic Games, and WTT Champions. Methods: Survival analysis was introduced, in which points scored in a match and shot counts in a rally were treated as "survival time," while losing a point was defined as the "event." The Kaplan-Meier method was employed to estimate cumulative survival probabilities, which reflect the likelihood of players enhancing their winning potential at specific scores or shot numbers. The point or shot number at which the cumulative survival probability drops to 50% is defined as the KP or KS, respectively. Additionally, the Log-rank (Mantel-Cox) non-parametric test was applied to determine whether significant differences existed between the survival curves of different groups. Results: KP in table tennis singles demonstrate a stable pattern of " 7 for women, 8 for men," while Winning Points (WP) are predominantly concentrated at the 9th point. The KS positions are consistent in both men's and women's singles, specifically the 4th shot in the total rally, the 3rd shot in the serving round, and the 2nd shot in the receiving round. Winning Shots (WS) consistently occur at the 6th shot in the total rally, as well as the 3rd shot in both the serving and receiving rounds. Conclusion: These findings advance our understanding of winning patterns in table tennis, providing theoretical foundations and practical references for designing contextualized training programs, precisely regulating athletes' competitive psychology, and optimizing future competition rules.

## Introduction

Sports performance analysis has emerged as an important subdiscipline within sports science [1,2], focusing on the systematic extraction of latent data from athletes' performance behaviors through multi-dimensional data collection and analysis.

**Data availability statement:** All relevant data are within the paper and its Supporting information files.

**Funding:** This study was supported by Social Science Foundation of Jiangsu Province (NO. 23TYD006).

**Competing interests:** The authors have declared that no competing interests exist.

Identifying key factors for success in sports competitions could provide coaches, athletes, and researchers with valuable messages for evaluating actual performance [3–5]. The critical role of performance analysis in sports is now globally acknowledged, particularly in the study of technical and tactical performance [6–8]. With technological advancements, the field has increasingly incorporated tools such as artificial intelligence and big data analytics, facilitating the generation of multi-dimensional, actionable insights. These insights not only refine athletes' technical skills and tactical strategies but also optimize coaches' decision-making processes and enhance the precision of technical adjustments.

Knowledge of the key factors influencing technical and tactical performance is essential for training and competition in numerous sports [9,10]. Research on these factors is generally divided into two primary categories: (1) Key Performance Indicator (*KPI*) analysis. For instance, in basketball, *KPI* analysis typically encompasses metrics such as fouls, steals, rebounds, two-point shots, and three-point shots. These indicators are analyzed about game-specific situations using advanced methodologies like machine learning and statistical modeling to identify the most critical *KPIs* under varying conditions [11,12]. Similarly, *KPI* analysis in table tennis focuses on metrics such as stroke position, stroke technique, stroke direction, and game phase, combined with situational variables, to highlight the *KPIs* of athletes during different phases and score ranges [13,14], Notably, Zhang et al. systematically reviewed match analysis methods in Chinese table tennis, emphasizing the "three-phase evaluation method" and its evolution into the "four-phase evaluation method," which segments matches into serve-attack, receive-attack, and stalemate phases to quantify technical effectiveness and winning probability [15]. This phased approach, enhanced by computer-aided and model-based analyses (e.g., Markov chains, neural networks), provides a structured framework for identifying KPIs and optimizing technical applications in specific competitive scenarios. (2) Key Points (*KP)* analysis. In tennis, for example, Roure C. have observed that players who win specific points, such as those at 30:30 or those who provide an initial lead, exhibit a significantly higher probability of winning the match. Those points were defined as "key points" [16]. Fitzpatrick et al. analyzed elite-level tennis matches on grass and clay surfaces, revealing that, for both men and women players, 0–4 points, first-serve points, and baseline points were strongly associated with match victories [17]. Similarly, in badminton, Wang C-C. applied binary entropy analysis to examine the relationship between scoring structures in individual matches and their outcomes, concluding that different scoring structures exert varying degrees of influence on match results [18]. As a racket sport integrating five key elements speed, power, placement, spin, and trajectory table tennis requires a multifaceted approach to performance analysis. Investigating the key determinants of success is crucial for elite athletes seeking to enhance their winning probability. However, the precise definitions of *KP* and *KS* during matches remain inadequately established, which to some extent hampers in-depth understanding and strategic optimization. Kayacan et al. conducted a systematic evaluation of Turkish table tennis leagues, correlating player profiles (e.g., victory or defeat, playing style, equipment, demographic data) with ranking points.

Their findings revealed that parameters such as match outcomes, nationality, and salary receipt significantly influenced rankings, whereas age, height, and equipment type showed no correlation [19]. This finding serves as a reminder that KPI are context dependent, making it imperative to adopt empirical research methods.

Currently, performance analysis in table tennis competitions primarily focuses on comparisons between athletes and the evaluation of technical performance indicators. Comparative studies often adopt a phased evaluation method to analyze the differences between Chinese athletes and their international counterparts [20–23]. Given the consistent dominance of Chinese table tennis teams in global rankings, such comparisons provide valuable insights into the relative strengths and weaknesses of athletes [24–26]. With advancements in computer technology, researchers have increasingly utilized sophisticated models, including Markov chains, association rule models, BP neural networks, and expert knowledge systems, to examine key technical and tactical indicators [13,27–32]. These analytical approaches facilitate the identification of critical factors influencing match outcomes at various stages, while also elucidating the important roles of technical actions and their combinations. Wang et al. developed a data mining-based analysis system that integrates multi-factor combinations (e.g., hitting technique, point, position) and employs both sequential and inverse correlation analysis modes to evaluate technical and tactical patterns [33]. Their system demonstrated high accuracy and efficiency in extracting actionable insights, providing coaches with a reliable tool for tactical decision-making and training optimization. Furthermore, studies focusing on stroke efficiency, game-related situational variables, line-changing strategies, and the impact of new plastic balls have enhanced the understanding of the determinants of success in table tennis from multiple perspectives [22,34–37]. However, the analysis of KP and KS in table tennis still mainly depends on the practical expertise of coaches and researchers, rather than systematic and evidence-based research reports.

This study pioneers the application of survival analysis to systematically investigate the KP and KS in singles matches among elite table tennis players. Survival models were developed based on score progression and shot sequences, enabling a comprehensive analysis across various match scenarios, including gender, strength difference (SD), match result, and game number scenarios. Given the diverse competitive scenarios inherent in table tennis, this study proposed the following initial hypotheses: (a) KP and KS would differ between male and female players, and (b) WP were likely to converge due to the comparable competitive levels among elite athletes. Guided by these hypotheses, the primary objective of this research was to precisely pinpoint KP and KS, thereby establishing a solid theoretical foundation for enhancing coaching strategies and optimizing athletes' tactical execution during competition.

## Methods

### Statistical Analysis

**(1) Statistical test.** The Log-rank (Mantel-Cox) non-parametric test was employed to compare the cumulative survival probabilities between groups in the sample. Under the null hypothesis, $H_0$ (i.e., the cumulative survival probabilities of the groups are equal) is valid, a comparison test is performed by contrasting the theoretical expected value with the observed value. It should be noted that the Log-rank test is specifically employed to determine whether significant differences exist between group survival curves, where a $p < 0.05$ indicates statistical significance and a $p < 0.01$ denotes high statistical significance.

All statistical analyses described above, including the calculation of cumulative survival probabilities and the Log-rank (Mantel-Cox) non-parametric test, were performed using SPSS 27.0 and GraphPad Prism 9.0 software.

**(2) Survival analysis.** The non-parametric Kaplan-Meier product-limit method was utilized in survival analysis to estimate cumulative survival probabilities (survival functions) and median survival times [38]. In the context of survival analysis, a positive event, such as a loss, occurring earlier or reaching the median survival time earlier corresponds to a lower survival probability [39]. In table tennis, two players alternate hitting the ball, with each point concluding when one player fails to return the ball, resulting in a point scored by the opponent. For survival analysis, a lost point is defined as a positive event. The scoring structure in table tennis comprises point, game, and match, with each game adhering to

 

an 11-point system. In this study, the losing player was designated as the observation target, and the entire game was treated as a survival process. The number of points scored by the losing player was regarded as the survival time. The median survival time of the losing player's points in a game was defined as "KP". In each point aspect, the loser of each point was considered the observer, and the process from the serve (first shot) to the conclusion of the point (last shot) was treated as a survival process. The total number of shots during this process was defined as the survival time. The median survival time of a single point, based on the total number of shots, was defined as the "KS".

The earlier the KP appears, the greater the probability that the player will lose the game. Similarly, the earlier the KS occurs, the higher the likelihood of losing a point. When an athlete's scores in a game reach the KP, or their shots at a point reach the KS, their performance surpasses 50% of the games, which means the winning probability of the game or the point would increase. Building on these concepts, this study introduces the Winning Point (WP) and Winning Shot (WS) as extensions of the key metrics. They are defined as the number of points or shots achieved by an athlete for the first time when the cumulative survival probability drops below 30%. Upon reaching the WP or WS, the athlete's performance exceeds 70% of the games, providing a significant advantage in securing a game or a point.

**(3) Calculation method.** The formula for calculating the cumulative proportion of surviving individuals was modified and applied based on the work of EL Kaplan et al. [38], as detailed below:

$$S(t) = P(T > t) \; ; \; t = 1, 2, 3, \ldots, n \tag{1}$$

$P(T > t)$ represents the probability of survival beyond a specific point or number of shots, where $t$ denotes the survival time corresponding to a particular point or shot count.

KP and KS are defined as the earliest survival time at which the cumulative survival probability drops to or below 0.5.

$$KP/KS = S(T \geq t) \leq 0.5 \tag{2}$$

WP and WS are defined as the earliest survival time at which the cumulative survival probability decreases to or below 0.3.

$$WP/WS = S(T \geq t) \leq 0.3 \tag{3}$$

## Match samples

This study selected a total of 124 international top-tier matches as the sample, comprising 60 men's singles and 64 women's singles matches, with participants including both left-handed and right-handed athletes. All matches are selected from competitions including the World Championships, World Cup, Olympic Games and WTT Champions, with participating athletes featured in the world rankings published by the International Table Tennis Federation (ITTF). Based on the ITTF world rankings and the final match score differences, matches were further categorized into Balanced, Unbalanced, and Blowout groups for finer analysis (see 'Classification of groups' for details). The total number of games played was 657 (327 for men and 330 for women), which included 12,161 points.

## Observed indicators

Based on the performance analysis framework introduced by Professor Tamaki S [10], a structured observation form was developed to capture multidimensional match data. There were 5 observation indicators in this study: Number of shots for each point, players who strike the ball in each shot, results of each point (score or loss), result per game (e.g., 9:11), result per match (4:0, 4:1, 4:2, 4:3).

## Classification of groups

Survival models in this study involved comparisons of several match scenarios, in the *KP* part, gender, strength difference (*SD*), match result, and game number scenarios included. The gender scenario included 2 groups: men's singles and women's singles. 3 groups were classified in the *SD* scenario based on the score differences between the two athletes: balanced group ($SD \leq 5$), unbalanced group ($6 < SD \leq 15$), and blowout group ($SD \geq 16$). *SD* is also the difference of game results, for example, if the result of a game is 11:9, the *SD* is 2, which equals 11 minus 9. Match result scenario included 4 groups: 4:0, 4:1, 4:2, and 4:3. Games number scenario included 7 groups: Game 1, Game 2, Game 3, Game 4, Game 5, Game 6, and Game 7. In the part of *KS*, there were 3 scenarios: gender, round, and match result. The classification of groups in gender and match result scenarios was the same as in the *KS* part. As to round scenarios, 3 groups were classified: Total round, serving round, and receiving round (Fig 1). In the total round, the shot number was counted for both players and when a player served, the shot number was recorded as 1, the opponent reserve, the shot number is recorded 2. Next, the player hits the 3rd shot and the opponent on the 4th shot until the point is lost. In the serving round and reserving round, the shot number was counted only according to the shots by the player in the specific round which is recorded as 1, 2, 3...

## Data collection

All match videos were obtained from publicly available sources, including television broadcasts and the official websites of the ITTF and WTT. All relevant data are provided in the manuscript and its Supporting Information files. To ensure the reliability of the data collection, two raters, who were professional players from the Chinese Table Tennis Association (CTTA), were recruited. They received standardized training in the data collection protocols and operational definitions of all indicators. Subsequently, to assess inter-rater reliability, both raters independently analyzed an identical set of 20

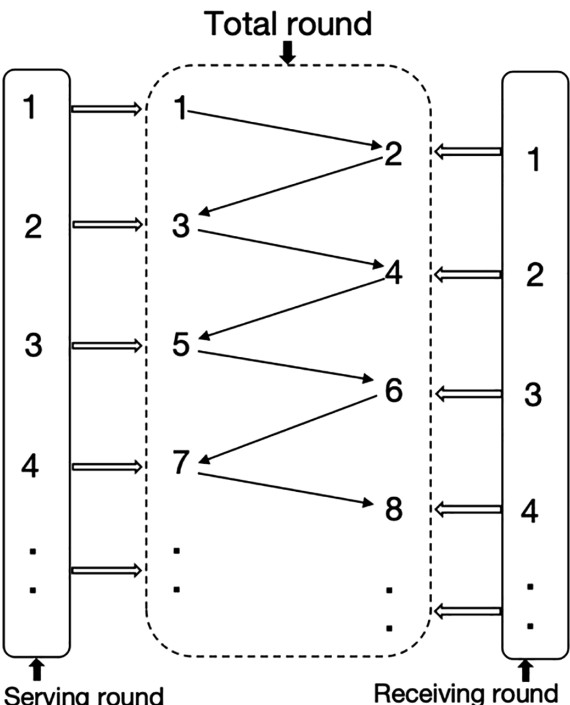

**Fig 1. Visual depiction of the classification of groups in round scenarios.** *Note*: 1, 2, 3, 4 et al mean shot numbers.

randomly selected matches. The consistency between their assessments was perfect, as indicated by a Cohen's kappa coefficient of $k = 1.00$.

## Results

### Comparison of *KP* in singles matches

First, a comparison of *KP* in men's and women's singles matches was conducted. The results, as shown in Fig 2, indicate no significant differences between men's and women's singles competitions ($p > 0.05$). The *KP* was 8 (S = 33%) in men's singles and 7 (S = 44%) in women's singles. In terms of the *WP*, the results show that the *WP* for men's singles was 9 (S=15%), while for women's singles, it was 8 (S=28%).

Next, the *KP* was analyzed between three groups within the *SD* scenario for both men's and women's singles matches. As depicted in Figs 3a and 3b, no significant differences were observed among the three groups in men's singles match

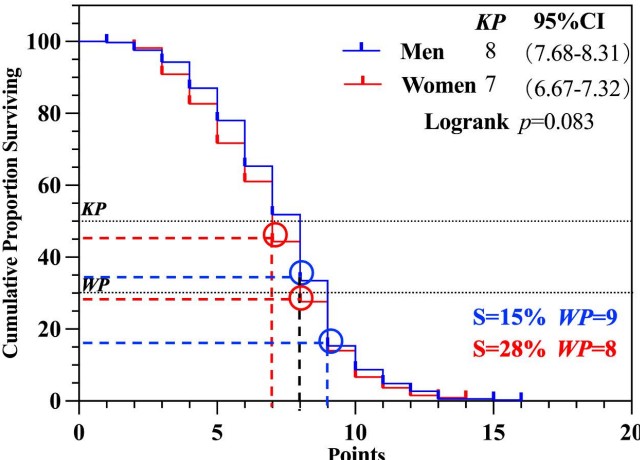

**Fig 2. *KP* and *WP* of gender scenario.**

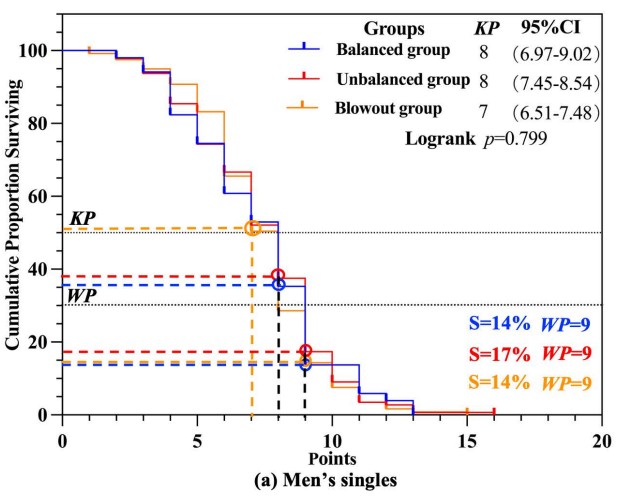
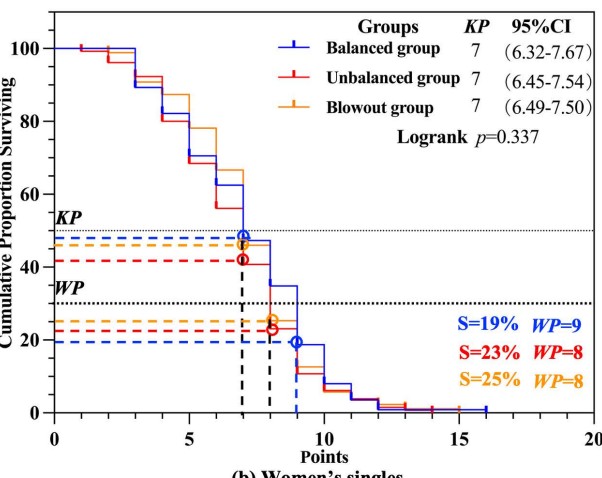

**Fig 3. *KP* and *WP* of strength difference scenario.**

($p>0.05$). Specifically, the *KP* for the balanced group was 8 (S=35%), for the unbalanced group, 8 (S=37%), and for the blowout group, 7 (S=50%). Similarly, no significant differences were identified in women's singles match ($p>0.05$), with a *KP* of 7 observed among all three groups: the balanced group (S = 47%), the unbalanced group (S = 41%), and the blowout group (S = 46%). The *WP* for the three men's groups ranged between 9 (S = 14% − 17%), whereas for the three women's groups, the *WP* ranged between 8 and 9 (S = 19% − 25%).

Survival analysis was conducted to investigate the *KP* in match result scenario for men's and women's singles competitions among four outcome groups. In table tennis singles matches, which follow a best-of-seven match, it is hypothesized that *KP* may vary depending on match outcomes. The results, illustrated in Figs 4a and 4b, indicated no significant differences among the four outcome groups in the men's singles match result scenario ($p>0.05$). Specifically, for 4:0 and 4:1 outcomes, the *KP* was consistently 7 (S = 47% − 48%), whereas for 4:2 and 4:3 outcomes, the *KP* was 8 (S = 37% − 41%). Similarly, in women's singles, no significant differences were observed among the four groups ($p>0.05$). The *KP* was consistently 7 for 4:0, 4:1, and 4:3 outcomes (S = 37% − 46%), while for the 4:2 outcome, the *KP* was 8 (S = 28%). Regarding the *WP*, the results showed that in men's singles, the *WP* was 8 (S = 22% − 29%) for 4:0 and 4:1 outcomes and 9 (S = 14% − 21%) for 4:2 and 4:3 outcomes. In women's singles, the *WP* across all four outcome groups was 8 (S = 24% − 30%).

Survival analysis was conducted to examine the *KP* in the game number scenario for men's and women's singles competitions. The results, presented in Fig 5a and 5b, showed no significant differences in *KP* in the game number scenario for both men's and women's singles ($p>0.05$). In men's singles, the *KP* for Games 1–3 was consistently 7 (S = 43.3% − 48.3%), while for Games 4–7, the *KP* was consistently 8 (S = 38.3% − 46.4%). In women's singles, the *KP* varied by game: for Game 1 and Game 6, the *KP* was 6 (S = 46.2% − 50%); for Game 2, it was 8 (S = 29.7%); for Games 3–5, it was 7 (S = 40.9% − 48.4%); and for Game 7, it was 5 (S = 25%). Regarding the *WP*, in men's singles, the *WP* for Games 1–6 was consistently 9 (S = 12% − 18%), while for Game 7, the *WP* increased to 10 (S = 13%). In women's singles, the *WP* was 8 (S = 15% − 30%) for Games 1, 2, 4, 5, and 6; 9 (S = 13%) for Game 3; and 10 (S = 25%) for Game 7.

## Comparison of *KS* in singles matches

In the gender scenario (Fig 6), no significant differences were observed in the total round between men's and women's singles matches ($p>0.05$), with the *KS* being all 4 (S = 47%).

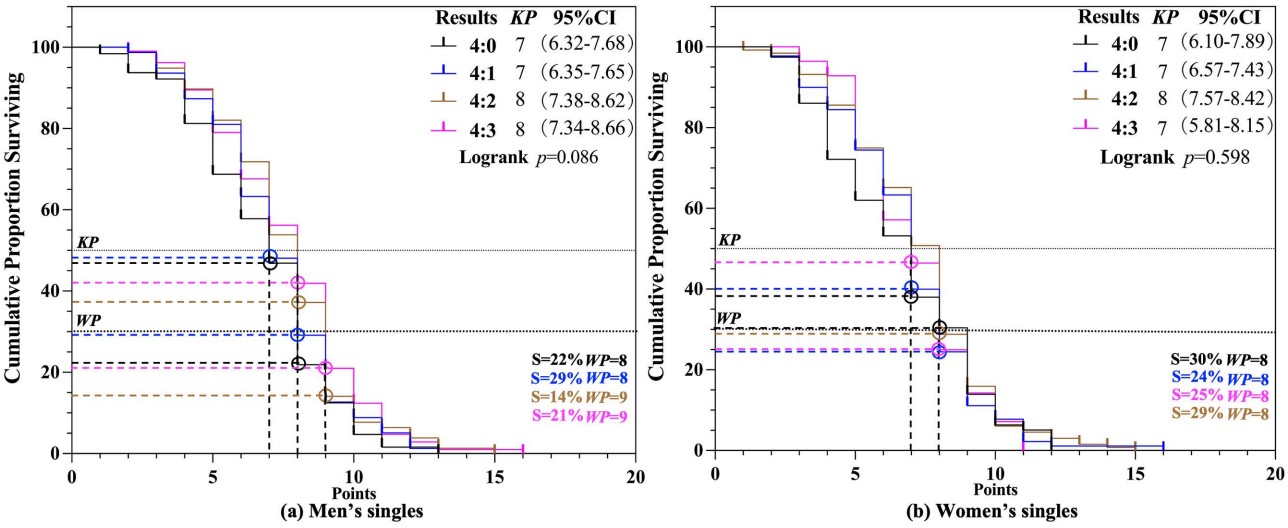

**Fig 4.** *KP* and *WP* of Match result scenario.

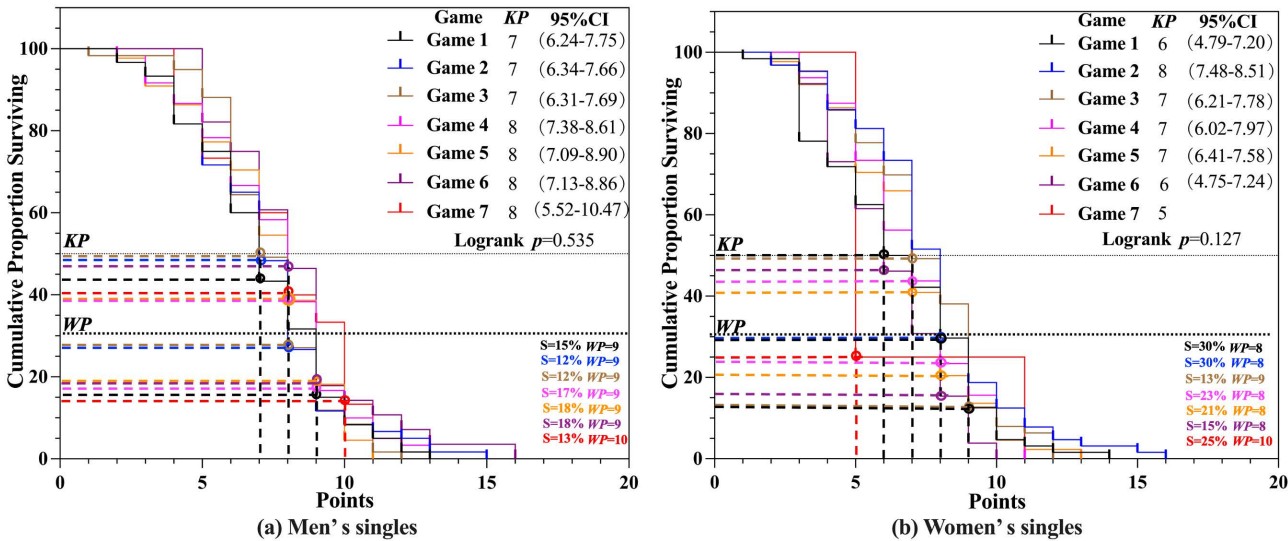

**Fig 5. *KP* and *WP* of Games number scenario.**

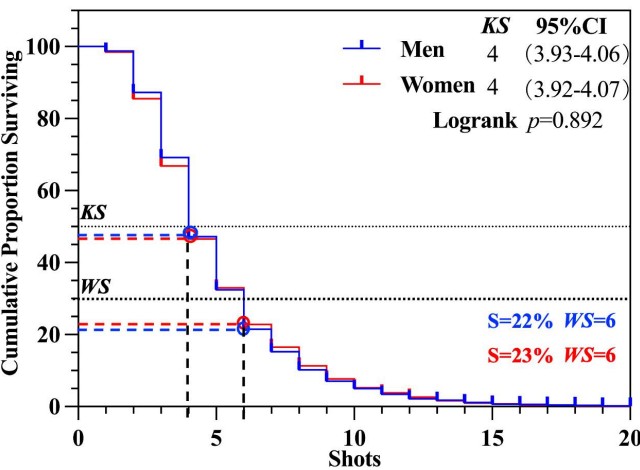

**Fig 6. *KS* and *WS* of gender scenario.**

Highly significant differences in *KS* were identified in men's singles matches in round scenario (Figs 7a and 7b), with similar result observed in women's singles matches ($p < 0.01$). For both men's and women's singles, the *KS* for the serving round was 3 (S = 26% − 28%), while for the receiving round, the *KS* was 2 (S = 38%). Regarding the *WS*, both men's and women's singles matches demonstrated a *WS* of 6 for the total round (S = 22% − 23%). For the serving round, the *WS* was consistently 3 (S = 26% − 28%), while for the receiving round, the *WS* was 3 (S = 17% − 19%).

In the Match Result scenario (Figs 8a and 8b), significant differences were observed in the total round among the four groups in men's singles ($p < 0.05$). Notable inter-group differences were identified between the 4:0 and 4:2 groups ($p < 0.05$) and between the 4:0 and 4:3 groups ($p < 0.01$). Despite these statistical differences, the *KS* remained

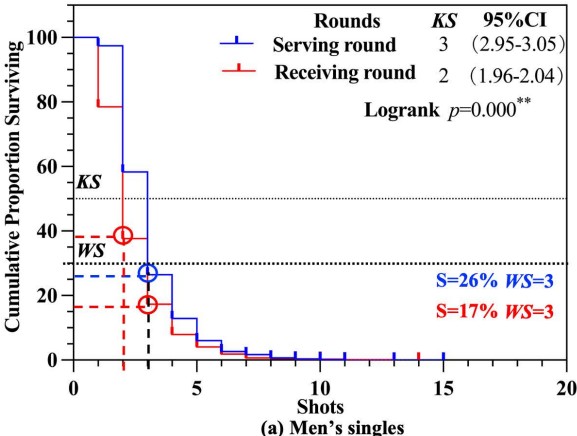

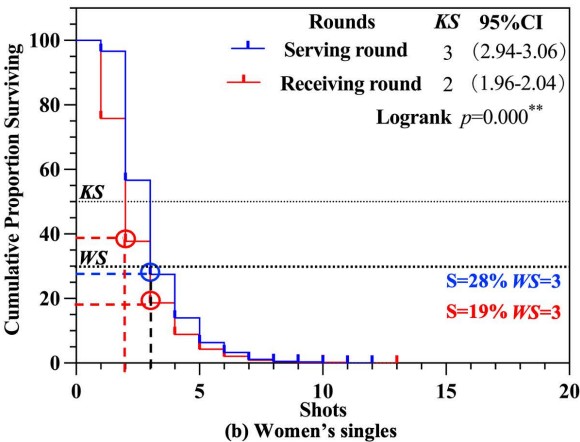

**Fig 7. KS and WS of round scenario.** *Note*:***, **, and * denote statistical significance at the 0.1, 1, and 5% levels, respectively.

numerically consistent across all four groups at the 4th shot (S = 44% − 49%). In women's singles, no significant differences were found in the total round ($p > 0.05$), with the KS remaining constant at the 4th shot (S = 45% − 47%) in all four groups. Additionally, the WS results showed that both men's and women's singles exhibited a WS of 6 (S = 20% − 24%) in different Match Result scenarios.

In the serving round (Figs 8c and 8d), no significant differences were observed in the KS of men's singles across the four groups ($p > 0.05$), with the KS remaining consistent at 3 (S = 23% − 28%) in all groups. In women's singles, however, significant differences were identified in the KS during the serving round ($p < 0.01$), with inter-group differences observed between the 4:3 and 4:0 groups, the 4:3 and 4:1 groups, and the 4:3 and 4:2 groups ($p < 0.01$). For the 4:0, 4:1, and 4:2 match result, the KS was 3 (S = 26% − 28%), while for the 4:3 result, the KS increased to 5 (S = 26%). Additionally, the WS results showed that, during the serving round, both men's and women's singles exhibited a WS of 3 (S = 26% − 29%).

In the receiving round (Figs 8e and 8f), no significant differences were observed in the KS between the four groups in men's singles ($p > 0.05$), with the KS remaining constant at 2 (S = 35% − 39%) across all groups. In women's singles, significant differences were identified in the KS during the receiving round ($p < 0.05$). For the 4:0, 4:1, and 4:2 match result, the KS was 2 (S = 37% − 38%), while for the 4:3 result, the KS increased to 4 (S = 37%). Significant inter-group differences were observed between the 4:3 and 4:0 groups, the 4:3 and 4:1 groups, and the 4:3 and 4:2 groups ($p < 0.01$). Regarding the WS, during the receiving round, the WS for men's singles was consistently 3 (S = 16% − 19%). In women's singles, the WS for the 4:0, 4:1, and 4:2 match result were 3 (S = 18% − 20%), while for the 4:3 result, the WS was 6 (S = 16% − 19%).

## Discussion

This study is the first to employ survival analysis to systematically investigate KP and KS in elite table tennis singles matches. Our findings reveal a robust gender-specific pattern in KP: "7 for women, 8 for men," with the WP consistently appearing at the 9th point. In terms of rally structure, KS consistently emerged at the 4th shot in the total rally, the 3rd shot in the serving round, and the 2nd shot in the receiving round. Meanwhile, WS were consistently observed at the 6th shot in the total rally and the 3rd shot in both serving and receiving rounds. These results provide statistically grounded benchmarks that transcend traditional experiential definitions of pivotal moments in table tennis.

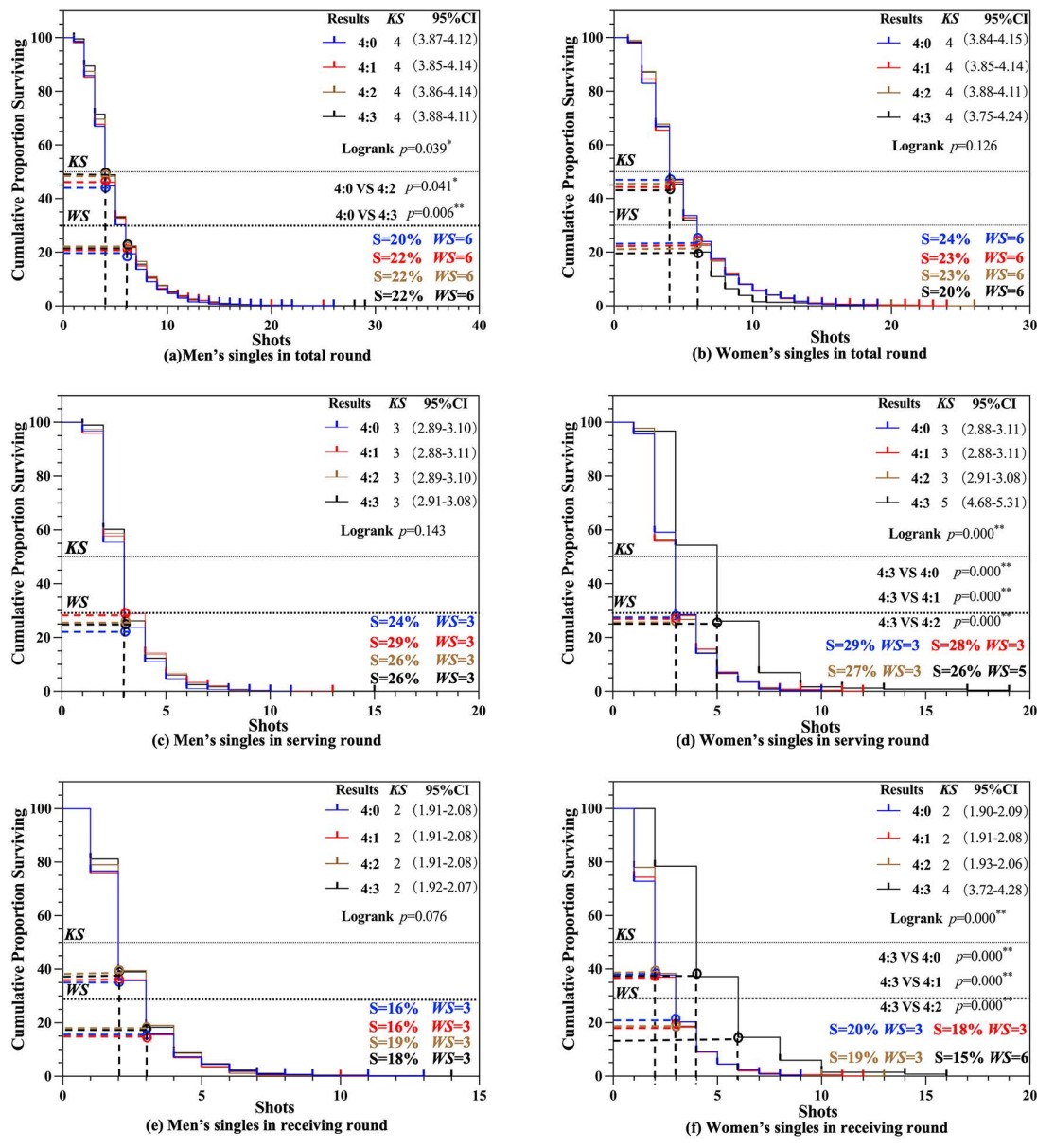

**Fig 8. *KS* and *WS* of Match Result scenario.**

## Key points and their significance in table tennis

The discussion on *KP* has been ongoing for a long time. Faming H suggested that *KP* typically refers to points where the score difference is within one or two points or where fluctuations occur during the middle stages of the game. He emphasized that positive emotions play an active and decisive role in key scores, while negative emotions can have a detrimental effect [40]. Similarly, Hong G et al. defined *KP* as those occurring after the score reaches 8:8 in a game [41]. Xinhui Zhao, however, argued that the *KP* in a match is the point occurring after the score reaches 9:9, as this point ultimately determines the outcome of the game [42]. Tao H et al. proposed a broader definition, identifying *KP* as any points scored when the combined scores of both players in a game are greater than or equal to 16 and the score difference is within 2

points (e.g., 9:7, 8:8, 7:9), continuing until the end of the game [43]. In practical terms, many scholars and coaches often classify the match into three periods: the initial period (1–4 points), the intermediate period (5–8 points), and the ending/crucial period (after 9 points). While these classifications provide a useful heuristic for understanding match dynamics, it is important to note that the analysis of *KP* in table tennis has historically been based on practical experience and intuition rather than systematic, data-driven research.

This study applied survival analysis to examine 124 table tennis singles matches. The results revealed no statistically significant differences in *KP* across various match scenarios, including gender, strength difference, match result, and number of games. The analysis confirmed that the *KP* in table tennis singles matches is "7 for women, 8 for men," while the winning point consistently occurs at the 9th point. The result means that when a male table tennis player scores 8 points or a female player scores 7 points in a singles match, his/her performance would generally outperform 50% of the single matches, while when a male/female table tennis player scores 9 points, his/her performance would outperform 70% of the single matches. The 11-point system in table tennis, where players must score 11 points to win a game, highlights the increasing significance of winning points as the score approaches 11 [34]. This underscores the significance of mastering the "7 for women, 8 for men" *KP* pattern to understand winning dynamics in table tennis. The insights from this study can be applied across various dimensions of the sport, including competition strategies, training, and potential reforms of game rules. (1) Helping Athletes Adjust Psychological State and Improve Tactical Execution. In elite-level matches, fluctuations in scores are often associated with changes in athletes' observable behaviors and performance outcomes, which may reflect underlying psychological shifts. Although not directly measured in this study, the consistent patterns at *KP* suggest these moments likely involve heightened psychological pressure, as supported by literature in racket sports [44,45]. Therefore, recognizing the timing and significance of *KP* may help athletes and coaches better anticipate match dynamics, regulate performance behaviors in real time, and execute tactics more effectively under competitive stress. This knowledge enhances their ability to maintain composure under pressure, improving overall performance during high-stakes moments. (2) Optimizing On-Site Competition Guidance by Leveraging Game Rules. Table tennis competition rules include provisions such as the "timeout" rule, which allows players to request a one-minute break, and the "towel break" rule, which permits athletes to wipe sweat after every six points. These rules can be strategically employed to optimize performance at key moments. For example, coaches and players can time these breaks to coincide with *KP*, helping players regain focus, reset strategies, and increase their chances of winning *KP*. Specifically, a timeout can be called when the score is approaching or at the *KP* (e.g., 7 or 8 points) to disrupt the opponent's rhythm and provide critical tactical guidance. Similarly, towel breaks can be utilized after the 6th point in a game, which often precedes the *KP*, allowing athletes to mentally prepare for the upcoming crucial phase. (3) Simulating *KP* Scenarios in Practice. Training programs can incorporate scenario-based drills that replicate *KP* situations, such as "8-point games" for male players or "7-point games" for female players, as well as specific score scenarios like "8:8" or "9:9". These simulations enhance athletes' ability to handle high-pressure situations, improve tactical stability during crucial moments, and develop psychological resilience. By practicing under these simulated conditions, athletes are better equipped to maintain composure and execute strategies effectively during real matches. Furthermore, training under pressure with specific scoring targets can improve decision-making accuracy and technical execution when it matters most. (4) Providing Reference for the Reform of Table Tennis Competition Rules. In recent years, the ITTF and World Table Tennis (WTT) have introduced several rule adjustments aimed at enhancing the spectator experience, attracting younger participants, and improving the sport's global competitiveness. The findings from this study, particularly the identification of *KP*, have the potential to inform future reforms by offering novel, data-driven insights to enrich the game. For instance: a) Inspired by the "2-point ball" rule in Ping Pong, a similar experimental rule could be trialed in WTT feeder or contender events around *KP*. Players could be granted one opportunity per match to request a "double-point rally" when the score is at or near the *KP* (e.g., at 7:7 in women's singles or 8:8 in men's singles). Winning this designated rally would award two points, instantly amplifying the stakes and excitement. This innovation could significantly enhance the drama and unpredictability of matches, while also creating pivotal

opportunities for athletes, particularly those from developing table tennis nations, to challenge higher-ranked opponents. b) Introducing a Strategic Timeout Before *KP*. Allowing a dedicated timeout for the receiving player just before a *KP* (e.g., when the score is 6:7 in women's or 7:8 in men's singles) would provide a critical opportunity to adjust strategies and mentally reset. This tactical interlude could heighten audience engagement by building suspense and enriching the narrative depth of the match. These findings could also guide decisions regarding the timing of commercial breaks during televised broadcasts, ensuring synchronization with critical moments to maximize audience engagement.

## Key shots and their significance in table tennis

Survival analysis of the *KS* in table tennis matches revealed that the *KS* remains largely consistent across various match scenarios, including gender, round, and match outcomes. In singles matches, the *KS* typically occurred at the fourth shot in the total round. In the serving round, the *KS* was observed at the third shot, while in the receiving round, it typically occurred at the second shot. However, in women's matches, the *KS* for the 4:3 winning score in the serving round extends to the fifth shot, and in the receiving round, it extends to the fourth shot. This finding may suggest that in closely contested women's matches, particularly those involving evenly matched opponents, the competition for each point is more intense, resulting in longer rallies and more shots per rally. The longer *KS* observed in women's matches during decisive games, without a corresponding increase in men's matches, may reflect differences in observed risk-coping behaviors under high-pressure situations. It is important to note that these interpretations are derived from behavioral performance data and existing theoretical frameworks, rather than direct measurement of psychological constructs. Women's play tends toward risk aversion: under the pressure of a decisive game, tactical execution shifts toward stability and control. Athletes prioritize minimizing unforced errors and prefer to engage in extended rallies, patiently maneuvering to accumulate advantages and capitalize on opponents' mistakes. This represents a rational risk-management strategy by prolonging the match. In contrast, men's play leans more toward risk-taking: even under high pressure, male athletes place greater reliance on the quality of their shots to actively score points. Their tactical execution focuses on initiating attacks early and applying aggressive power, aiming to establish dominance within the first few shots or directly end the rally. Consequently, the number of *KS* does not increase significantly, reflecting a risk-management approach centered on ending the point decisively. In summary, this disparity does not reflect a difference in ability levels but rather represents two rational risk-decision models based on distinct competitive characteristics, thereby illustrating the rich diversity of tactical execution in the sport of table tennis. Additionally, *WS* consistently occurred at the sixth shot in the total round and is all on the third shot in the serving round and receiving round.

In racket sports, the outcome of a point typically involves one or more shots of counterplay. Research on the significance of shots has given rise to two predominant perspectives. One perspective posits that the final shot plays the most decisive role in determining the match outcome [46,47], with the influence of shot counts becoming increasingly pronounced as the match progresses. Based on this view, Liu et al proposed a new model of shot performance relevance (SPR) [34]. The second perspective emphasizes the importance of the initial shots in a game. Several studies have underscored the critical role of the first three to four shots in table tennis. Furthermore, following multiple revisions of table tennis competition rules, and reductions in speed and spin that have led to longer rallies, Tong L. and Qing Y. observed that the fifth shot has become increasingly critical under the new rules compared to earlier periods [47,48].

The rules of table tennis stipulate that, in singles matches, players alternate shots until a point is scored or lost. Each point consists of multiple shots, making the identification of *KS* within a rally crucial for understanding the dynamics of the sport. (1) Absolute and Relative Importance. Table tennis matches follow a structured temporal sequence in which players alternate shots. Early shots, such as serve and receive, inherently carry absolute importance, as failure in these shots prevents the continuation of subsequent shots. Similarly, the final shot, which directly determines the outcome of the point, also holds absolute significance. However, no study has yet defined the typical position of the final shot in the sequence. The importance of *KS* lies within this framework. This study employs survival analysis to examine the shot number for over 50%

of final shots (those resulting in a lost point), offering insights into the relative importance of specific shots. For example, if most players do not score or lose points during the serve or receive, the absolute importance of these shots remains, but their relative significance diminishes. Conversely, if a specific shot is frequently associated with scoring or losing points, it highlights a universally challenging aspect of the game. By identifying *KS*, this study illuminates common challenges within the shot sequence of a table tennis rally. (2) Identifying *KS* helps to gain a better understanding of the patterns in table tennis. In China, the classic "three-phase evaluation theory" divides a table tennis match into three key phases [46]. On this basis, Yang and Zhang proposed the "four-phase evaluation method" by dividing the points scored and points lost on the fifth shot into different phases [47], and now is widely used in practice [15,24]. This type of scientific research is invaluable for improving the competitive abilities of table tennis players. Different phases and shots require specialized technical skills. By clarifying players' strengths and weaknesses in these areas, training can be more effectively tailored, leading to enhanced performance during matches. Specifically, coaches can design drills focusing on the 3rd shot in the serving round and the 2nd shot in the receiving round to strengthen athletes' capabilities at the *KS*. Moreover, for women's matches that go to a deciding game, training should emphasize sustaining rally length beyond the 4th or 5th shot to adapt to the extended *KS* under pressure. (3) Contribution to the Reform of Table Tennis Competition Rules. The *KS* findings suggest that table tennis singles matches primarily concentrate on the first five shots. Compared to other racket sports, such as badminton, tennis, and volleyball, rallies in table tennis remain relatively short. Increasing rally length, delaying the appearance of *KS*, and enhancing the overall excitement of the game represent potential areas for consideration by the ITTF in future rule reforms. A potential measure is to limit players to a single serve per point during the latter phase of a game (e.g., after 8 total points have been scored). This adjustment would increase pressure on the server, promote a more balanced dynamic between serving and receiving, and ultimately contribute to the later appearance of the *KS*.

## Limitation

First, while survival analysis is effective for identifying *KP* and *KS*, the model inherently simplifies complex match dynamics. It treats all players and rallies as homogeneous processes, potentially failing to fully capture the complexities introduced by player-specific factors such as playing style, handedness, world ranking, or tactical preferences. Second, this study did not incorporate real-time psychological or physiological data (e.g., emotional state, heart rate, or perceived pressure). Consequently, the interpretation of athletes' psychological states during *KP* and *KS* situations, as discussed, is based on the observation of match behaviors, existing literature, and theoretical deduction, rather than empirical measurement using tools such as psychological scales. This data gap limits the depth of understanding regarding the underlying mechanisms of athletes' decision-making and performance under pressure. Finally, as the findings are derived from singles matches, whose dynamics fundamentally differ from those of doubles and mixed doubles involving four interacting players, the generalizability of the results to these formats requires further validation. Future research should incorporate player-specific variables, integrate psychophysiological measurements, and extend the analytical framework to doubles disciplines to provide more comprehensive insights.

## Conclusion

This study represents the inaugural application of survival analysis to investigate *KP* and *KS* in singles matches of elite table tennis players. The findings reveal that key points in singles matches exhibited a consistent pattern of "7 for women, 8 for men" with winning points predominantly occurring at the 9th point. In both men's and women's singles, the *KS* were consistent, with the 4th shot in the total round, the 3rd shot in the serving round, and the 2nd shot in the receiving round. *WS* consistently occurred at the 6th shot in the total round and were all at the 3rd shot in the serving round and receiving round. The findings can guide the development of match strategies, inform training plans, enhance on-site decision-making, and support psychological adjustments during competitions and provide references for the future development of table tennis competition rules.

## Supporting information

**S1 Data. Supporting information—data.**
(XLSX)

## Acknowledgments

None.

## Author contributions

**Conceptualization:** Muzi Li.

**Data curation:** Muzi Li, Qing Yang.

**Formal analysis:** Muzi Li.

**Funding acquisition:** Muzi Li, Qing Yang.

**Writing – original draft:** Muzi Li, Qing Yang.

**Writing – review & editing:** Muzi Li, Qing Yang.

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
