## [Decision Letter · Decision Letter 0]

2 Sep 2025

Dear Dr. Yang,

Thank you for submitting your manuscript to PLOS ONE. After careful consideration, we feel that it has merit but does not fully meet PLOS ONE’s publication criteria as it currently stands. Therefore, we invite you to submit a revised version of the manuscript that addresses the points raised during the review process.

We look forward to receiving your revised manuscript.

Kind regards,

Veysel Temel

Academic Editor

PLOS ONE

Journal Requirements:

2. In your Methods section, please include additional information about your dataset and ensure that you have included a statement specifying whether the collection and analysis method complied with the terms and conditions for the source of the data.

“This study was supported by Social Science Foundation of Jiangsu Province (NO. 23TYD006).”

4. We note that your Data Availability Statement is currently as follows: All relevant data are within the manuscript and in Supporting Information files.

6. Please ensure that you refer to Figure 1 in your text as, if accepted, production will need this reference to link the reader to the figure.

Additional Editor Comments:

Reviewer’s 1 Comments to the Authors

Dear Authors,

Thank you for submitting your manuscript entitled “Exploring the key points and key shots in table tennis matches based on survival analysis” to PLOS ONE. The manuscript addresses an important and timely topic, providing valuable insights into key points and key shots in elite table tennis matches. After careful review, the manuscript has been evaluated positively, with minor revisions recommended to enhance clarity, contextualization, and methodological description.

Based on the comments provided by Reviewer 1 (Dr. Yücel Makaracı), the following points should be addressed in your revised manuscript:

1. Abstract

o Clarify the type and level of matches analyzed (e.g., tournament, championship).

o Provide a brief explanation of survival analysis for readers unfamiliar with this method.

o Refine the conclusion to directly reflect the study findings.

2. Introduction

o Correct any typographical errors and clearly identify the referenced researchers.

o Ensure the discussion focuses on table tennis rather than other sports.

o Provide references to support claims and clarify which comparative studies are mentioned.

o Clearly state the study hypotheses.

o Consider incorporating the suggested references to strengthen context and discussion.

3. Materials and Methods

o Provide detailed information regarding match selection criteria, type, and level.

o Re-examine specific lines for accuracy as highlighted by the reviewer.

o Consider restructuring the Methods section by introducing a Statistical Analyses subsection first, followed by detailed description of survival analysis and other methods.

4. Results

o Presentation is generally clear; minor adjustments may be made for clarity if necessary.

5. Discussion

o Begin with a short summary paragraph of the main results.

o Acknowledge limitations regarding psychological and tactical interpretations of key points.

o Discuss possible reasons for observed differences in key shots, particularly in women’s matches, including technical, tactical, and psychological factors.

o Expand the limitations section to consider player-specific factors, lack of physiological or psychological data, and simplifications inherent in survival analysis.

Reviwer’s 2 Editor’s Comments to the Authors

Dear Authors,

Thank you for submitting your manuscript entitled “Exploring the key points and key shots in table tennis matches based on survival analysis” to PLOS ONE. The study addresses an important topic and provides valuable insights into key points and key shots in elite table tennis matches. Following the review process, the manuscript has been evaluated positively, but minor to moderate revisions are required to enhance clarity, methodological transparency, and contextualization.

Based on the reviewers’ comments, please address the following points in your revised manuscript:

1. Abstract

• Clarify the type, level, and selection criteria of matches analyzed.

• Specify the statistical methods used, including survival analysis and any tests applied.

• Provide a brief explanation of survival analysis for readers unfamiliar with the method.

• Refine the conclusion to directly reflect the study findings.

2. Introduction

• Correct any typographical errors and clearly identify referenced researchers.

• Ensure the focus remains on table tennis rather than other sports.

• Provide references to support all claims and clarify which comparative studies are mentioned.

• Clearly state the study hypotheses.

• Consider incorporating the suggested references to strengthen context and discussion.

3. Materials and Methods

• Provide detailed information regarding match selection criteria, type, level, and age group of participants.

• Describe whether athlete-specific characteristics (e.g., handedness, ranking, playing style) were considered.

• Explain how observation forms were created and whether expert support was obtained.

• Specify how cumulative survival probabilities were calculated and whether expert validation was applied.

• Clearly attribute the source of any formulas included in the Methods.

• Explain the rationale for using specific statistical tests, such as the Log-Rank nonparametric test, including the meaning of reported p-values (e.g., 0.05, 0.01).

• Consider restructuring the Methods section with a Statistical Analyses subsection for clarity.

4. Results

• Ensure all statistical results are fully described, including how p-values were calculated.

• Tables and figures should be clear and well-labeled.

5. Discussion

• Begin with a short summary of main results.

• Discuss psychological aspects cautiously: clarify how athletes’ psychological states were determined (e.g., via scales or observation), and acknowledge limitations if no direct measurements were obtained.

• Provide explanations for observed differences in key shots, considering technical, tactical, and psychological factors.

• Expand the limitations section to include player-specific factors, lack of physiological/psychological data, and simplifications inherent in survival analysis.

6. Suggestions and Practical Implications

• Revise the suggestions section to provide more specific, actionable recommendations for players, coaches, and future research.

Academic Editor’s Comments to the Authors

Dear Authors,

Thank you for submitting your manuscript entitled “Exploring the key points and key shots in table tennis matches based on survival analysis” to PLOS ONE. The manuscript addresses an important and timely topic, and both reviewers acknowledge the value and rigor of your study.

After careful consideration of the reviewers’ comments, I recommend that the manuscript undergo minor to moderate revisions before it can be considered for publication. The reviewers have highlighted several areas requiring clarification and elaboration, particularly regarding:

1. Methodological details: Please clarify match selection criteria, participant characteristics (e.g., age group, handedness, ranking), observation form development, and statistical procedures including the survival analysis and Log-Rank tests. Ensure that all formulas and calculations are properly sourced and explained.

2. Abstract and Introduction: Provide a brief explanation of survival analysis for readers unfamiliar with the method, specify the type and level of matches analyzed, refine the conclusion to directly reflect study findings, and strengthen the introduction with appropriate references and clear study hypotheses.

3. Results and Statistical Reporting: Ensure that all statistical results, including p-values and significance thresholds, are clearly reported and described.

4. Discussion and Interpretation: Address the discussion of psychological states cautiously, clearly stating how these were measured or observed. Consider expanding the limitations to acknowledge factors such as player-specific differences, absence of physiological or psychological measurements, and simplifications inherent in survival analysis.

5. Practical Implications: Revise the suggestions section to provide actionable recommendations for players, coaches, and future research.

The reviewers acknowledge that the manuscript is generally well-written and presents its findings clearly. Addressing these points will strengthen the manuscript’s clarity, methodological transparency, and overall contribution to the field.

We look forward to receiving a revised manuscript that comprehensively addresses these comments.

Sincerely,

Academic Editor

PLOS ONE

Reviewers' comments:

Reviewer's Responses to Questions

**Comments to the Author**

1. Is the manuscript technically sound, and do the data support the conclusions?

Reviewer #1: Yes

Reviewer #2: Yes

2. Has the statistical analysis been performed appropriately and rigorously?

Reviewer #1: Yes

Reviewer #2: Yes

3. Have the authors made all data underlying the findings in their manuscript fully available?

Reviewer #1: Yes

Reviewer #2: Yes

4. Is the manuscript presented in an intelligible fashion and written in standard English?

Reviewer #1: Yes

Reviewer #2: Yes

Reviewer #1: Abstract

The statistical method used is not stated in the abstract or method.

Method:

-How were the selected matches selected?

-What criteria did you use while watching?

-For example, was it considered whether the athlete was right-handed or left-handed?

-The statistical method used is not stated in the abstract or method.

-Which age group was the study conducted for? And where was it conducted?

-How were the observation forms created? Was expert support obtained?

-How was cumulative survival probability calculated? Was it calculated by an expert?

-There are formulas in the method section, whose formulas are these? No source.

-What is the score of the Log-Rank nonparametric test? Why is it nonparametric?

According to the above review, the method should be corrected again.

Findings

-The findings include 0.05 and 0.01. How was this calculated? It should be stated.

Discussion

-Why were psychological states included in the discussion without adhering to a specific criterion?

-The psychological states of the athletes were mentioned in the discussion. How were the athletes' psychological states determined? Was a scale used?

Suggestions

-The suggestions are insufficient; they should be rewritten.

Reviewer #2: Dear Authors,

First, I would like to commend you for your efforts in conducting this important study. The manuscript presents a clearly described investigation into the key points and key shots in singles matches for elite table tennis players, enhancing our understanding of winning patterns in the sport. While the study provides valuable insights, there are several areas that require clarification or revision. In order for the manuscript to be considered for publication, I kindly ask that you address the following comments.

Abstract

L33: Please clarify which matches were analyzed (e.g., tournament, championship, competition level).

L34: Readers may not be familiar with survival analysis. Please provide a brief definition or explanation, while considering the word limit.

L42–45: Refine the conclusion so that it more directly reflects the current findings.

Introduction

The introduction requires further development to fully explain the research context. Specific issues include:

L72: Please double-check for a possible typo.

L72: Specify which researchers are being referred to.

L78: The discussion shifts to badminton. After presenting other sports, the focus should return to table tennis.

L82: Provide a reference to support this claim.

L83: Define which “comparative studies” are mentioned.

L104: Clearly state the study hypotheses.

I recommend incorporating the following references to strengthen the introduction and discussion:

DOI: 10.1007/s00500-023-09082-z

DOI: 10.15640/jpesm.v2n2a12

DOI: 10.1080/02640414.2018.1460050

Materials and Methods

L106: Provide more detail about which matches were analyzed (e.g., type, level, and selection criteria).

L127: Please re-check this line for accuracy.

L143: Consider restructuring the methods section by introducing a “Statistical Analyses” subsection first, and then describing survival analysis and other methods.

Results

The results and tables are generally well-presented. Congratulations on this clear presentation.

Discussion

The discussion is well-structured, but several points require attention:

L286: Begin the discussion with a short paragraph summarizing the main results before introducing sub-sections.

The emphasis on the psychological role of key points (KP) is strong, but there is no empirical evidence on how emotions or tactics actually shift at KP. Please acknowledge this gap.

The connection between KP thresholds and psychological/tactical execution remains hypothetical; this should be noted as a limitation.

Women’s matches are observed to have longer key shots (KS) in close games, but the reasons (technical, tactical, psychological) are not sufficiently explored.

The limitations section could be expanded to acknowledge:

Player-specific factors (e.g., style, handedness, ranking, tactical preferences).

Lack of psychological or physiological data (e.g., emotional state, heart rate, perceived pressure).

Simplifications of match dynamics inherent in survival analysis.

.

Reviewer #1: No

Reviewer #2: No

While revising your submission, please upload your figure files to the Preflight Analysis and Conversion Engine (PACE) digital diagnostic tool, https://pacev2.apexcovantage.com/. PACE helps ensure that figures meet PLOS requirements. To use PACE, you must first register as a user. Registration is free. Then, login and navigate to the UPLOAD tab, where you will find detailed instructions on how to use the tool. If you encounter any issues or have any questions when using PACE, please email PLOS at . PACE helps ensure that figures meet PLOS requirements. To use PACE, you must first register as a user. Registration is free. Then, login and navigate to the UPLOAD tab, where you will find detailed instructions on how to use the tool. If you encounter any issues or have any questions when using PACE, please email PLOS at . PACE helps ensure that figures meet PLOS requirements. To use PACE, you must first register as a user. Registration is free. Then, login and navigate to the UPLOAD tab, where you will find detailed instructions on how to use the tool. If you encounter any issues or have any questions when using PACE, please email PLOS at . PACE helps ensure that figures meet PLOS requirements. To use PACE, you must first register as a user. Registration is free. Then, login and navigate to the UPLOAD tab, where you will find detailed instructions on how to use the tool. If you encounter any issues or have any questions when using PACE, please email PLOS at figures@plos.org. Please note that Supporting Information files do not need this step.. Please note that Supporting Information files do not need this step.

---

## [Author Response · Author response to Decision Letter 1]

31 Oct 2025

Dear reviewers，

Thank you very much for your valuable comments and suggestions. We appreciate and are greatly motivated by the kind comments that recognize the potential of our work. We have done our best to address all the concerns raised and revised the paper accordingly.

We have updated the cover letter, supplemented the raw data, and provided point-by-point responses to the reviewers' comments. We hope that this revised version addresses all the concerns of the reviewers. Due to the large amount of content in this revision, if you approve our revision, we will invite native English speakers to polish the full text. Your assistance in reviewing this paper is highly appreciated.

Yours sincerely,

Qing Yang, Mu-zi Li.

---

## [Decision Letter · Decision Letter 1]

20 Nov 2025

Exploring the key points and key shots in table tennis matches based on survival analysis

PONE-D-25-36951R1

Dear Dr. Yang,

We’re pleased to inform you that your manuscript has been judged scientifically suitable for publication and will be formally accepted for publication once it meets all outstanding technical requirements.

Kind regards,

Veysel Temel

Academic Editor

PLOS ONE

Additional Editor Comments (optional):

Thank you for the opportunity to revise and resubmit our manuscript entitled “Exploring the Key Points and Key Shots in Table Tennis Matches Based on Survival Analysis.” We appreciate the constructive and insightful feedback provided by the reviewers. Their comments substantially improved the clarity, methodological rigor, and interpretative depth of the study.

All reviewer critiques were addressed comprehensively in the revised version. Methodological explanations were expanded, statistical procedures were clarified, additional robustness checks were incorporated where requested, and the presentation of results was strengthened to ensure greater transparency and interpretability. We also made extensive editorial improvements to enhance the overall coherence of the manuscript.

We confirm that the revised manuscript reflects all required corrections and is now, in our view, substantially stronger and more complete. We thank you for overseeing the review process and for the time and effort dedicated by both you and the reviewers. We hope that the revised manuscript meets the journal’s standards and is now suitable for publication.

Reviewers' comments:

Reviewer's Responses to Questions

**Comments to the Author**

Reviewer #3: (No Response)

Reviewer #4: All comments have been addressed

2. Is the manuscript technically sound, and do the data support the conclusions?

Reviewer #3: Yes

Reviewer #4: Yes

3. Has the statistical analysis been performed appropriately and rigorously?

Reviewer #3: Yes

Reviewer #4: Yes

4. Have the authors made all data underlying the findings in their manuscript fully available?

Reviewer #3: Yes

Reviewer #4: Yes

5. Is the manuscript presented in an intelligible fashion and written in standard English?

Reviewer #3: Yes

Reviewer #4: Yes

Reviewer #3: (No Response)

Reviewer #4: Dear authors,

Thank you for revising the manuscript and providing detailed responses. I am pleased to see that all the comments and suggestions raised in the previous review have been addressed appropriately. The revisions have notably enhanced the clarity, coherence, and overall structure of the manuscript. In my view, the article in its current form is suitable for publication.

.

Reviewer #3: No

Reviewer #4: No

---

## [Editor Report · Acceptance letter]

PONE-D-25-36951R1

PLOS One

Dear Dr. Yang,

I'm pleased to inform you that your manuscript has been deemed suitable for publication in PLOS One. Congratulations! Your manuscript is now being handed over to our production team.

Kind regards,

on behalf of

Professor Veysel Temel

Academic Editor

PLOS One